# Evaluation of Melanocyte Loss in Mycosis Fungoides Using SOX10 Immunohistochemistry

**Cynthia Reyes Barron** and **Bruce R. Smoller** *

Department of Pathology and Laboratory Medicine, University of Rochester Medical Center, Rochester, NY 14642, USA; creyesba711@gmail.com
* Correspondence: bruce_smoller@urmc.rochester.edu

**Abstract:** Mycosis fungoides (MF) is a subtype of primary cutaneous T-cell lymphoma (CTCL) with an indolent course that rarely progresses. Histologically, the lesions display a superficial lymphocytic infiltrate with epidermotropism of neoplastic T-cells. Hypopigmented MF is a rare variant that presents with hypopigmented lesions and is more likely to affect young patients. The etiology of the hypopigmentation is unclear. The aim of this study was to assess melanocyte loss in MF through immunohistochemistry (IHC) with SOX10. Twenty cases were evaluated, including seven of the hypopigmented subtype. The neoplastic epidermotropic infiltrate consisted predominantly of CD4+ T-cells in 65% of cases; CD8+ T-cells were present in moderate to abundant numbers in most cases. SOX10 IHC showed a decrease or focal complete loss of melanocytes in 50% of the cases. The predominant neoplastic cell type (CD4+/CD8+), age, race, gender, histologic features, and reported clinical pigmentation of the lesions were not predictive of melanocyte loss. A significant loss of melanocytes was observed in 43% of hypopigmented cases and 54% of conventional cases. Additional studies will increase our understanding of the relationship between observed pigmentation and the loss of melanocytes in MF.

**Keywords:** mycosis fungoides; hypopigmented mycosis fungoides; SOX10; cutaneous T-cell lymphoma



## 1. Introduction

Mycosis fungoides (MF) is the most common subtype of primary cutaneous T-cell lymphoma and is characterized by the presence of malignant T-cells within the epidermis (epidermotropism) [1]. The neoplastic T-cells often have a phenotype that is CD3 positive, CD4 positive, and CD8 negative with a frequent loss of other T-cell markers such as CD5 and/or CD7 [2]. Clonal T-cell receptor (TCR) gamma gene rearrangements, that may be determined by PCR-based amplification and gel electrophoresis, support the diagnosis [2]. MF usually follows an indolent course but may progress from a patch stage to plaque to tumor and disseminated disease [3]. There is generally good clinicopathologic correlation for these stages of disease and they display distinct findings histologically [4]. Although mortality is low, MF may undergo large cell transformation with significant reduction in survival [5].

MF is a disease primarily affecting adults, and patients typically present with persistent erythematous patches and plaques in skin that is sun protected such as the trunk (see Figure 1). However, the presentation may be quite variable and non-specific with a clinical differential including psoriasis and eczema, among others. Multiple biopsies are often required for diagnosis. Reflectance confocal microscopy, a non-invasive method of visualizing the skin in horizontal planes, may be a useful tool for initial screening in the clinic because the findings correlate well with histologic findings including the identification of epidermotropic lymphocytes [6]. Dermoscopy for the evaluation of inflammatory lesions is also becoming more common and frequent findings in MF include the presence of white scales and predominantly patchy vessel arrangements [7]. The clinical findings,

including features observed using confocal microscopy and dermoscopy, may help guide the decision to biopsy.

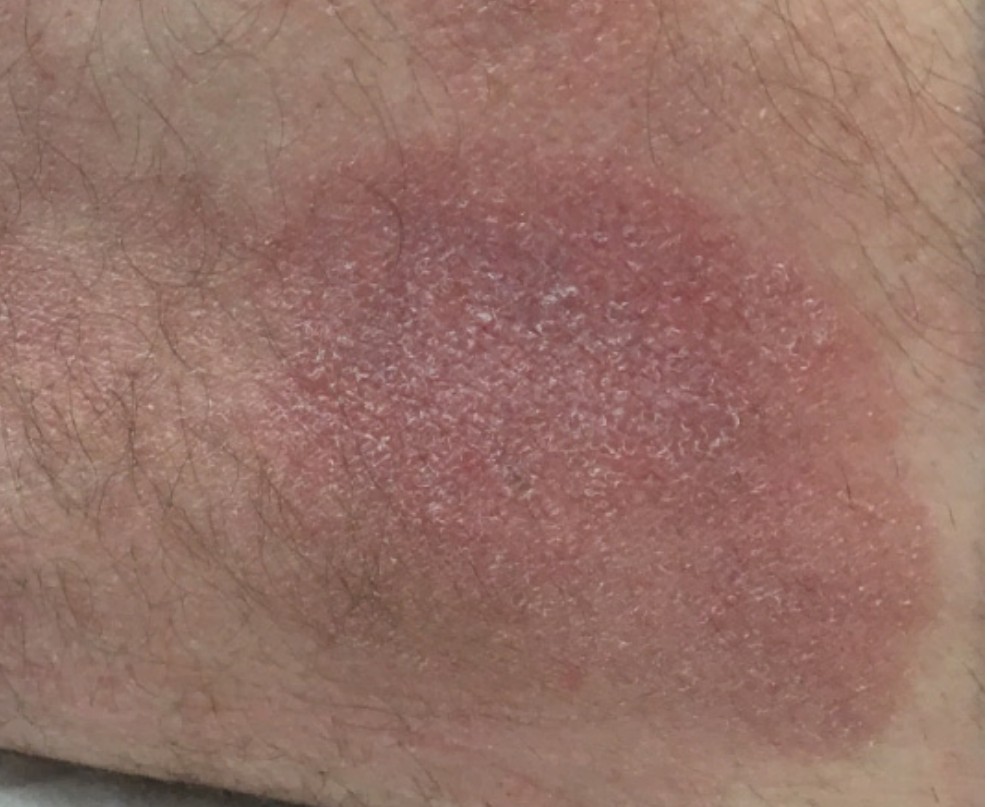

**Figure 1.** Mycosis fungoides presenting as a scaly, erythematous patch in a Caucasian patient.

There are numerous subtypes of MF with different clinical presentations adding to the diagnostic challenge. These subtypes include pagetoid reticulosis, folliculotropic, poikilodermatous, bullous, granulomatous, and hypopigmented [8]. Hypopigmented MF is a rare variant that is more likely than other types of cutaneous T-cell lymphoma to affect young patients and patients of African American or Hispanic descent [9,10]. A recent study of Chinese patients found that the age of patients with hypopigmented MF was significantly younger than for other variants with a median of 13.4 years and these patients had a good response to treatment [11]. Clinically, patients present with patches or plaques in sun-protected areas, as in other types of MF, but the lesions have lighter coloration than the surrounding skin (see Figure 2). There may also be a mixed presentation with the presence of both hypopigmented and erythematous lesions [11]. The clinical differential diagnosis is broad and includes vitiligo, pityriasis alba, post-inflammatory hypopigmentation, and leprosy [8]. Biopsy and histopathologic analysis are imperative in making the diagnosis.

SOX10 is a protein that functions as a transcription factor that is essential to the differentiation of neural crest cells. One of its many functions involves promoting the development and survival of melanocytes by regulating the expression of multiple genes [12]. An immunohistochemical (IHC) stain with an antibody to the SOX10 protein is widely available in many laboratories and is an important tool for the evaluation of melanocytic lesions. SOX10 strongly stains the nuclei of melanocytes including melanocytes in benign nevi and melanoma [13]. The absence of staining indicates the absence of melanocytes as is seen in depigmented lesions of vitiligo. SOX10 may be a helpful stain in further understanding the etiology of the loss of pigment in hypopigmented MF. Preliminary studies have shown that other melanocyte IHC markers including Melan-A, tyrosinase, stem cell factor receptor (CD117), and microphthalmia-associated transcription factor (MiTF) appear to be diminished in hypopigmented MF [14]. The sensitivity and specificity of

SOX10 for melanocytes may make it a useful stain to further study the pathophysiology of hypopigmentation in this disease. The diagnosis of hypopigmented MF requires clinical observation. Determining whether hypopigmentation results from the loss of the melanin pigment only or true loss of melanocytes will require an assessment of lesions from both conventional MF and the rare hypopigmented subtype.

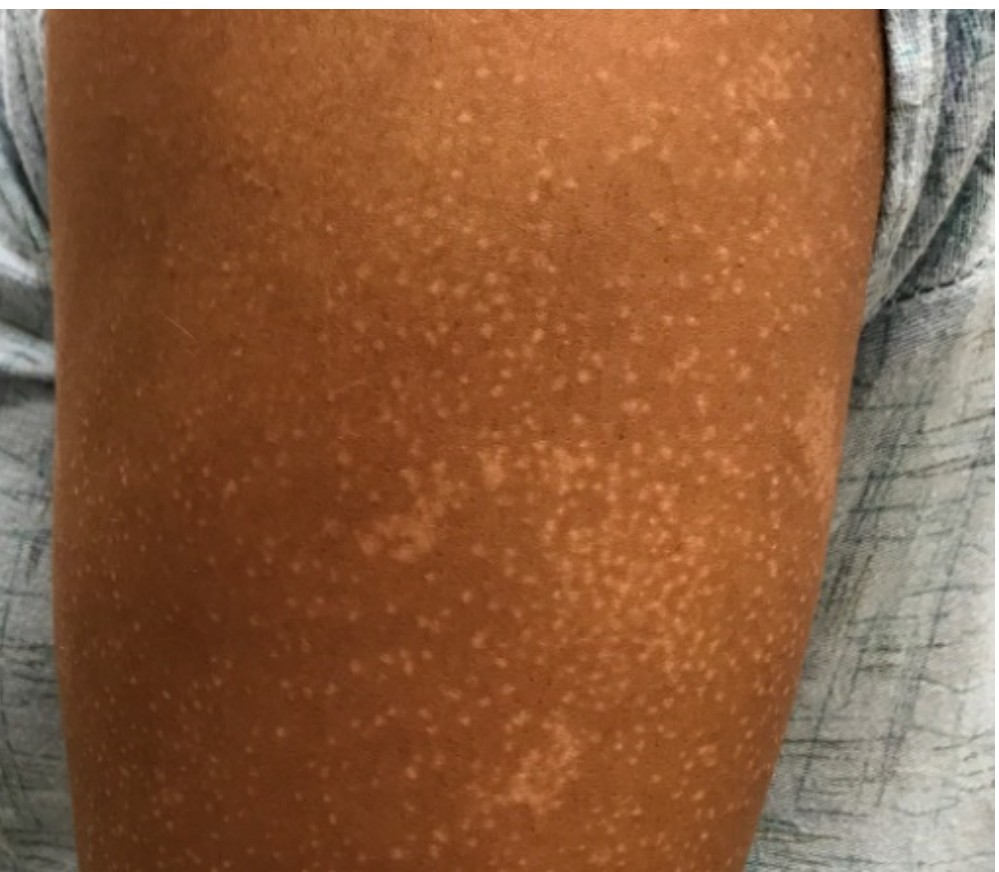

**Figure 2.** Mycosis fungoides presenting as hypopigmented macules and patches in the upper extremity of an African American patient.

## 2. Materials and Methods

A search was conducted to identify cases of MF with a clinical presentation of hypopigmentation in the pathology archives at the Department of Pathology and Laboratory Medicine of the University of Rochester Medical Center. Seven cases that were hypopigmented clinically were identified from January 2010 to March 2020. An additional 13 cases of conventional MF were selected from the same period to include cases from a wide age range and cases where the malignant T-cells were predominantly CD8+. The histopathologic findings for each case were reviewed including previously performed IHC studies for CD4 and CD8. All cases but one had previously reported clonal T-cell receptor gamma gene rearrangements assessed by PCR and gel electrophoresis. The clinical presentation as well as patient gender, age, and ethnicity were reviewed for each case.

The paraffin tissue blocks were retrieved from the archives for the selected cases and new tissue sections were cut. SOX10 IHC was performed to evaluate melanocyte loss using the mouse monoclonal IgG antibody BC34 from Biocare (1:200 dilution) on the Leica BOND stainer with Bond Polymer Refine Red detection kit. The antibody had been previously validated for clinical use. The staining results were assessed using standard light microscopy. Appropriate positive controls were confirmed as well as internal controls for each case. Cases were classified as having melanocyte loss if there was an absence of SOX10 staining in at least one contiguous 2 mm section of epidermis.

## 3. Results

Only seven cases with clinically diagnosed hypopigmentation were identified in a ten-year period. The average patient age of these cases was 43 years (range 30 to 62). The majority of the hypopigmented cases were female (71%). The ethnicity of 86% of the hypopigmented cases was non-Caucasian and only 15% of the conventional cases. Of the hypopigmented cases, 57% were CD4 predominant (see Table 1).

**Table 1.** Analysis of 20 cases of mycosis fungoides with features observed in hypopigmented and conventional cases.

| Results | Hypopigmented | Conventional | All |
|---|---|---|---|
| Melanocyte loss | 3 (43%) | 7 (54%) | 10 (50%) |
| CD4 predominant | 4 (57%) | 9 (69%) | 13 (65%) |
| Ethnicity non-Caucasian | 6 (86%) | 2 (15%) | 8 (40%) |
| Gender (female) | 5 (71%) | 4 (31%) | 9 (45%) |
| Age (Average) | 43 | 54 | 50 |
| Range | 30–62 | 15–86 | 15–86 |

An abnormal SOX10 staining pattern indicating melanocyte loss was observed in 10 cases (50%). Seven of these cases were conventional MF and had no clinically observed hypopigmentation; however, in four of these, the epidermotropic malignant population of T-cells was predominantly CD8+ (see Figure 3).

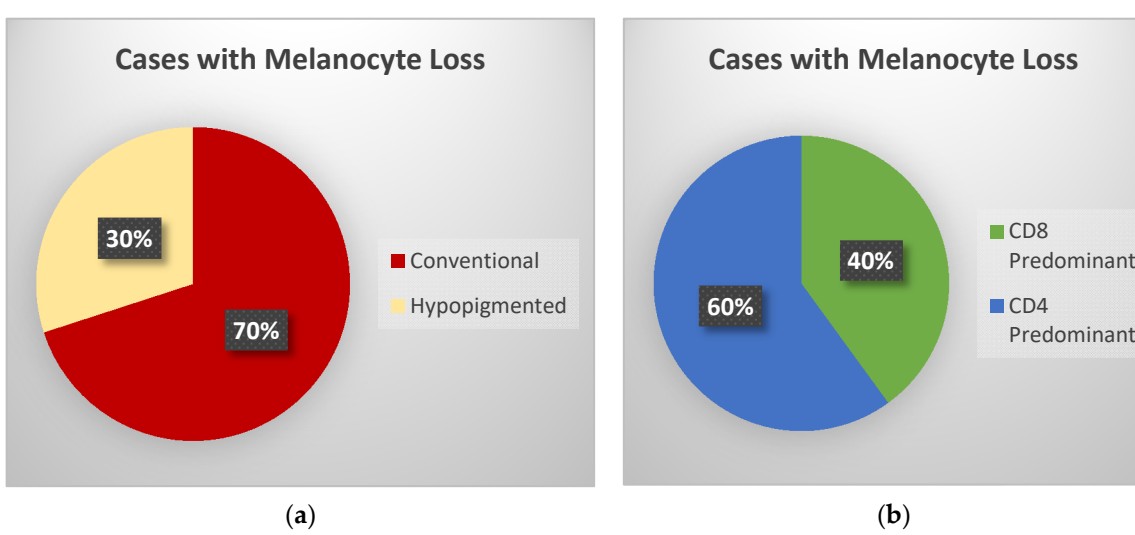

(**a**)　　　　　　　　　　　　　(**b**)

**Figure 3.** Ten cases displayed melanocyte loss by SOX10 IHC. The loss was observed in both conventional and hypopigmented cases (**a**) and in CD8 or CD4 predominant cases (**b**).

The remaining three cases that were predominantly CD8+ displayed melanocytes in normal quantities and distribution by SOX10 IHC (see Figure 4). Of the thirteen CD4 predominant cases, six had melanocyte loss by SOX10 IHC (see Figure 5).

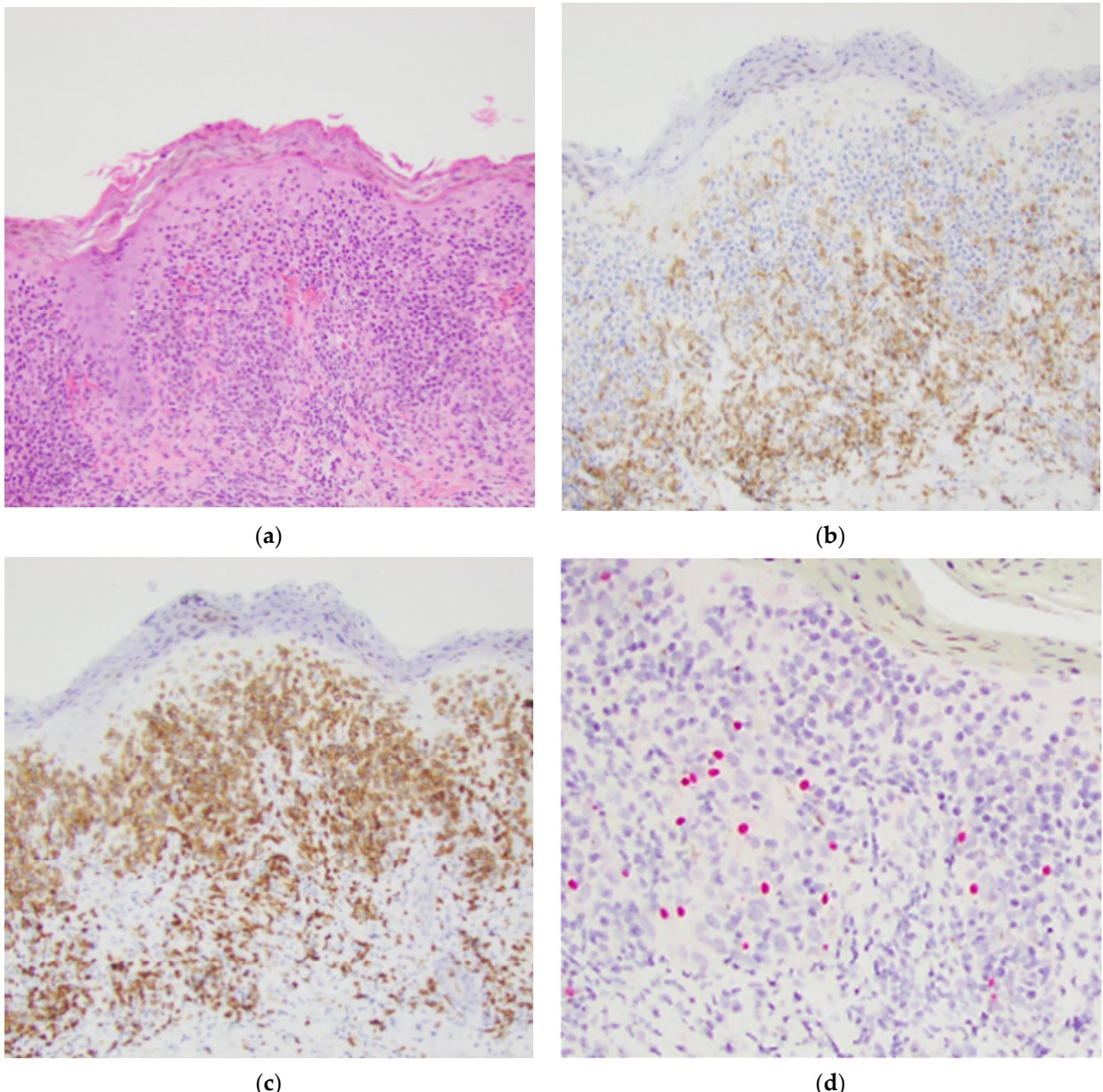

**Figure 4.** Sample case with a predominant CD8+ epidermotropic T-cell infiltrate with melanocytes in normal quantities and distribution highlighted by SOX10 IHC. The patient was a 40-year-old male with persistent pink to hyperpigmented patches in sun-protected areas. (**a**) Hematoxylin and eosin stained section, original magnification 100×. (**b**) T-cells highlighted by a CD4 IHC stain, original magnification 100×. (**c**) T-cells highlighted by a CD8 IHC stain, original magnification 100×. (**d**) SOX10 IHC highlighting melanocytes, original magnification 100×.

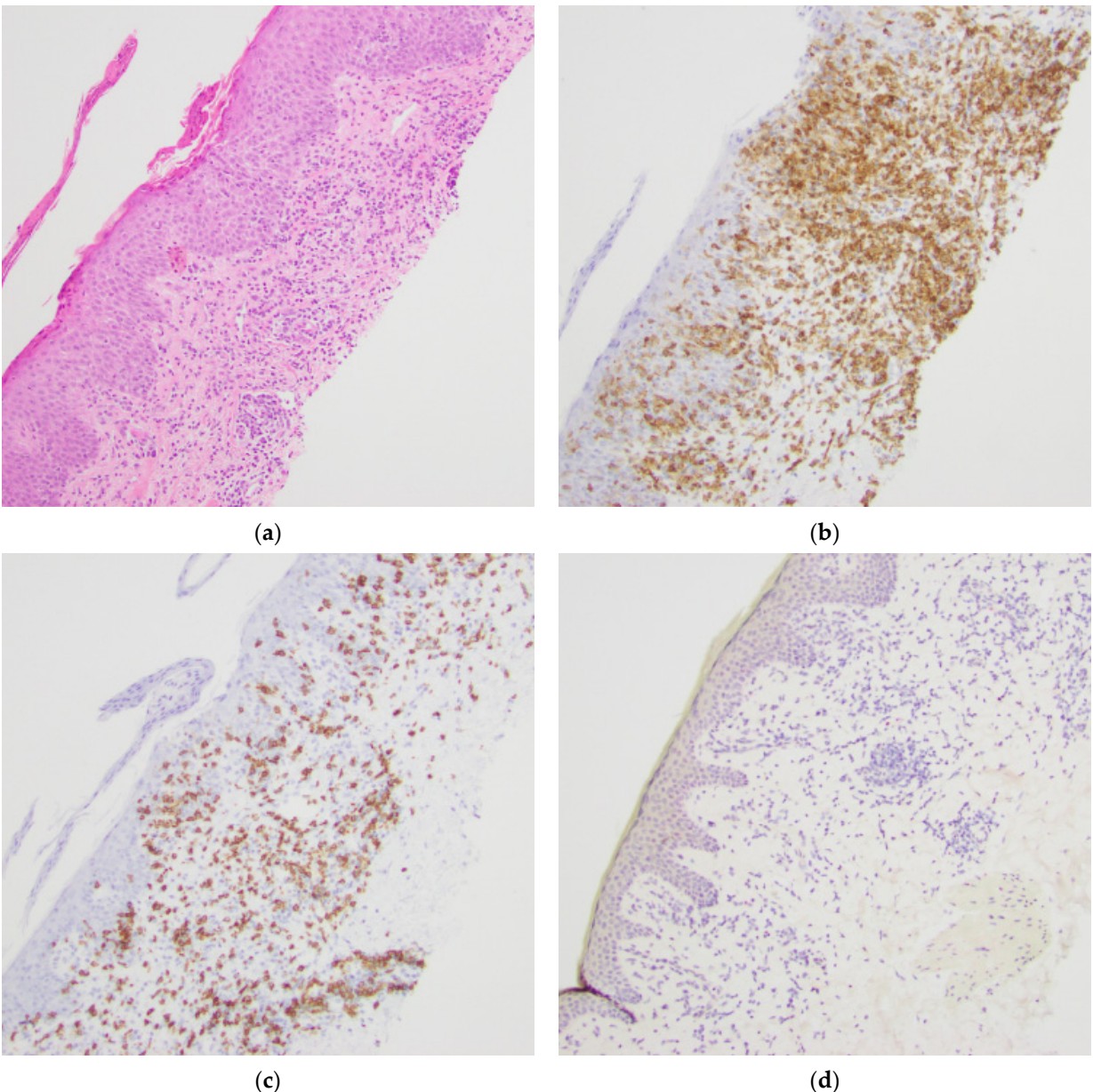

**Figure 5.** Sample case with a predominant CD4+ epidermotropic T-cell infiltrate with melanocyte loss shown by complete focal loss of SOX10 IHC staining. The patient was an 86-year-old male with a long history of hyperpigmented rash in sun-protected areas. (**a**) Hematoxylin and eosin-stained section, original magnification 100×. (**b**) T-cells highlighted by a CD4 IHC stain, original magnification 100×. (**c**) T-cells highlighted by a CD8 IHC stain, original magnification 100×. (**d**) Negative SOX10 IHC staining, original magnification 100×.

## 4. Discussion

The diagnosis of MF depends on the synthesis of pertinent histopathologic, molecular, and clinical findings [15]. The clinicopathological correlation is imperative. Hypopigmented MF is a rare subtype classified as such by clinical exam. Hypopigmentation is not merely an interesting clinical observation but designates the disease as a specific entity with better prognosis than other variants. Moreover, repigmentation of lesions may indicate effective treatment response and remission while recurrent hypopigmentation may indicate relapse [10]. Hypopigmented MF may have features that distinguish it from other variants of MF histopathologically as well, including the abundance of pigmentary incontinence [4] and melanocyte loss [14]. Some have argued that it is a separate disease from conventional MF altogether because the neoplastic T-cells are more frequently CD8+ cytotoxic T-cells and

the disease affects younger patients. Although hypopigmented MF generally has a better prognosis than conventional MF, rare cases of disease progression have been reported [16]. In this study, the average age was younger (43 years versus 54) and all but one case occurred in non-Caucasian patients including African Americans and South Asians. The population of malignant T-cells was CD4+ not CD8+ in the majority of the hypopigmented cases (57%). However, moderate to abundant numbers of CD8+ cytotoxic T-cells were present in the inflammatory infiltrate whether they were the malignant T-cell population or not.

Recent studies have shown that CD8+ cytotoxic T-cells play an important role in disease regulation in conventional MF when they form part of the antitumor immune response [17]. Tumor infiltrating lymphocytes (TILs) and the cytotoxic molecules they produce may be distinct from the molecules produced by the malignant T-cell population in MF. For example, granzyme B may be expressed in TILs but not in malignant cells [17]. Tumor necrosis factor alpha (TNFα) may be another important cytokine with high levels in stable lesions. Cases of CTCL disease progression or unmasking in previously undiagnosed patients who take TNFα inhibitors have been reported illustrating the potential critical role of TNFα in the immunologic control of MF and other types of CTCL [18,19].

The damage to melanocytes in cases of conventional and hypopigmented MF as shown by decreased SOX10 IHC staining in this study may be the result of collateral damage caused by the host immune response to malignancy and not damage caused directly by the malignant population of T-cells. Although hypopigmented MF may be more frequently seen in tumors with neoplastic cells that express CD8, we observed hypopigmentation in cases in which the neoplastic cells express CD4, as well. Likewise, other studies have reported a significant number of hypopigmented MF with malignant T-cells of the CD4+ phenotype indicating that CD8 positivity is insufficient for predicting the clinical findings of hypopigmentation and histologic melanocyte damage [17]. Patient age, race, and gender were not sufficient to predict melanocyte loss by SOX10 IHC.

Using hypopigmentation as a marker of active immune response has been proposed [17]. The clinical observation of hypopigmentation may be difficult in patients with fair skin tones such as Fitzpatrick types I and II. Furthermore, pigment loss may be clinically obscured by erythema from inflammation, trauma from scratching, background skin pigmentation, and partial treatment. Histologically, distinguishing between malignant T-cells and T-cells that are part of the tumor response is very difficult and grading the extent of tumor infiltrating lymphocytes in MF is not feasible. Although cytologic atypia may be identified, an attempt to distinguish atypical malignant T-cells from reactive T-cells in the tumor response histologically is unlikely to be accurate. SOX10 IHC, as a sensitive and specific method of assessing melanocyte loss, has the potential to be a surrogate marker for tumor immune response. It may provide the ability to categorize MF as the hypopigmented variant or a variant with better prognosis by histopathological means. Additional studies with a greater number of cases having substantial clinical information would be helpful in determining the utility of SOX10 IHC in further categorizing MF.

**Author Contributions:** All authors contributed to conceptualization, methodology, formal analysis, and investigation. Writing of the original draft was performed by C.R.B. Supervision and project administration were conducted by B.R.S. Both authors have read and agreed to the published version of the manuscript.

**Funding:** This research received no external funding.

**Institutional Review Board Statement:** Not applicable.

**Informed Consent Statement:** Not applicable.

**Acknowledgments:** Special thanks to Sierra Kovar for the technical support.

**Conflicts of Interest:** The authors declare no conflict of interest.

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
