# Peer review of "Evaluation of Melanocyte Loss in Mycosis Fungoides Using SOX10 Immunohistochemistry"

_dermatopathology, doi:10.3390/dermatopathology8030034_

Round 1

Reviewer 1 Report

The authors present a well-written concise report.

To further improve the article, please consider discussing other previous works e.g.:

  • Rodney IJ, Kindred C, Angra K, Qutub ON, Villanueva AR, Halder RM. Hypopigmented mycosis fungoides: a retrospective clinicohistopathologic study. J Eur Acad Dermatol Venereol. 2017 May;31(5):808-814. doi: 10.1111/jdv.13843. Epub 2016 Aug 12. PMID: 27515575.
  • Domínguez-Gómez, M.A., Baldassarri-Ortego, L.F. and Morales-Sánchez, M.A. (2021), Hypopigmented mycosis fungoides: A 48-case retrospective series. Australas J Dermatol. https://doi.org/10.1111/ajd.13565
  • Luo Y, Liu Z, Liu J, Liu Y, Zhang W, Zhang Y. Mycosis Fungoides and Variants of Mycosis Fungoides: A Retrospective Study of 93 Patients in a Chinese Population at a Single Center. Ann Dermatol. 2020 Feb;32(1):14-20. doi: 10.5021/ad.2020.32.1.14. Epub 2019 Dec 27. PMID: 33911704; PMCID: PMC7992633.

Please also add information on the manufacturer or the Sox10 antibody used.

Author Response

Comments and Suggestions for Authors

The authors present a well-written concise report.

To further improve the article, please consider discussing other previous works e.g.:

Rodney IJ, Kindred C, Angra K, Qutub ON, Villanueva AR, Halder RM. Hypopigmented mycosis fungoides: a retrospective clinicohistopathologic study. J Eur Acad Dermatol Venereol. 2017 May;31(5):808-814. doi: 10.1111/jdv.13843. Epub 2016 Aug 12. PMID: 27515575.

Domínguez-Gómez, M.A., Baldassarri-Ortego, L.F. and Morales-Sánchez, M.A. (2021), Hypopigmented mycosis fungoides: A 48-case retrospective series. Australas J Dermatol. https://doi.org/10.1111/ajd.13565

Luo Y, Liu Z, Liu J, Liu Y, Zhang W, Zhang Y. Mycosis Fungoides and Variants of Mycosis Fungoides: A Retrospective Study of 93 Patients in a Chinese Population at a Single Center. Ann Dermatol. 2020 Feb;32(1):14-20. doi: 10.5021/ad.2020.32.1.14. Epub 2019 Dec 27. PMID: 33911704; PMCID: PMC7992633.

Thank you for suggesting these excellent references.  The findings of the Rodney paper and Luo paper have now been incorporated into the introduction and discussion.  However, we do not have access to the Dominguez-Gomez paper.

Please also add information on the manufacturer or the Sox10 antibody used.

The SOX10 antibody was the BC34 clone from Biocare and additional information on the clone used was added to the second paragraph of the Materials and Methods section.

Reviewer 2 Report

The authors investigated the melanocyte loss in MF by SOX10 immunostaining on a series of 20 cases of MF. including 7 cases of hypopigmented MF. They found a significant loss of melanocytes in approximately 43% of hypopigmented MF cases and 54% of classic-type MF cases.

The paper is well written and sounds interesting.

I have two minor concerns:

  1. In the introduction, some details on the clinical diagnosis (dermoscopy and confocal-based) of MF should be added. For example the following doi should be considered: 10.1111/cup.13384.
  2. The authors, if possibile, should expand the cohort of MF cases examined.

Author Response

Comments and Suggestions for Authors

The authors investigated the melanocyte loss in MF by SOX10 immunostaining on a series of 20 cases of MF. including 7 cases of hypopigmented MF. They found a significant loss of melanocytes in approximately 43% of hypopigmented MF cases and 54% of classic-type MF cases.

The paper is well written and sounds interesting.

I have two minor concerns:

In the introduction, some details on the clinical diagnosis (dermoscopy and confocal-based) of MF should be added. For example the following doi should be considered: 10.1111/cup.13384.

Thank you for this suggestion.  We have added dermoscopy and confocal microscopy findings in MF in the second paragraph of the introduction and two additional references.

The authors, if possible, should expand the cohort of MF cases examined.

Unfortunately, adding new cases is not feasible for the first author at this time.  We hope that publishing our work with a limited number of cases, that illustrate a spectrum of presentations, will increase interest in the area and encourage others to investigate the loss of melanocytes in MF and add their findings to the published literature.